# Yeast Expressed Hybrid Peptide CLP Abridged Pro-Inflammatory Cytokine Levels by Endotoxin Neutralization

**DOI:** 10.3390/microorganisms11010131

**Published:** 2023-01-04

**Authors:** Junhao Cheng, Baseer Ahmad, Muhammad Asif Raza, Henan Guo, Marhaba Ahmat, Xubiao Wei, Lulu Zhang, Zhongxuan Li, Qiang Cheng, Jing Zhang, Junyong Wang, Dayong Si, Yueping Zhang, Rijun Zhang

**Affiliations:** 1Laboratory of Feed Biotechnology, State Key Laboratory of Animal Nutrition, College of Animal Science and Technology, College of Veterinary Medicine, China Agricultural University, Beijing 100193, China; 2Faculty of Veterinary and Animal Sciences, Muhammad Nawaz Shareef University of Agriculture, Multan 2500, Pakistan; 3School of Pharmaceutical Sciences, Tsinghua University, Beijing 100193, China; 4Tsinghua-Peking Center for Life Sciences, Beijing 100193, China; 5College of Bioengineering, Sichuan University of Science & Engineering, Chengdu 610017, China

**Keywords:** hybrid peptide, endotoxin, cytotoxicity, inflammation, immunomodulatory, apoptosis

## Abstract

The aim of this study was to apply a strategy to express a recombinant CLP peptide and explore its application as a product derived from natural compounds. The amphiphilic CLP peptide was hybridized from three parent peptides (CM4, LL37, and TP5) and was considered to have potent endotoxin-neutralizing activity with minimal cytotoxic and hemolytic activity. To achieve high secretion expression, an expression vector of pPICZαA-*HSA-CLP* was constructed by the golden gate cloning strategy before being transformed into *Pichia pastoris* and integrated into the genome. The recombinant CLP was purified through the Ni-NTA affinity chromatography and analyzed by SDS-PAGE and mass spectrometry. The Limulus amebocyte lysate (LAL) test exhibited that the hybrid peptide CLP inhibited lipopolysaccharides (LPS) in a dose-dependent manner and was significantly (*p* < 0.05) more efficient compared to the parent peptides. In addition, it essentially diminished (*p* < 0.05) the levels of nitric oxide and pro-inflammatory cytokines (including TNF-α, IL6, and IL-1β) in LPS-induced mouse RAW264.7 macrophages. As an attendant to the control and the parental peptide LL37, the number of LPS-induced apoptotic cells was diminished compared to the control parental peptide LL37 (*p* < 0.05) with the treatment of CLP. Consequently, we concluded that the hybrid peptide CLP might be used as a therapeutic agent.

## 1. Introduction

Lipopolysaccharide (LPS) or endotoxin is released into the immediate environment, afterward killing the Gram-negative bacteria. It is one of the imperative constituents that induce inflammation [1]. The portion of LPS (lipid A) is recognized by TLR4, which activates the inflammatory response that ultimately leads to sepsis [2]. Antibiotics have been commonly used in the treatment of inflammation, but they have many side effects [3]. Antibiotics can upsurge bacterial LPS by killing bacteria and activating the immune system to secrete cytokines (such as IL-6, IL-1β, and TNF-α) and produce toxic shock. Consequently, there is an urgent need to develop novel anti-inflammatory agents that have both LPS-neutralizing and antibacterial activities. Antimicrobial peptides (AMPs) widely exist in animals, plants, and microorganisms and are a kind of evolutionarily conserved molecule with a significant antibacterial activity that plays an indispensable part against pathogenic microbial infections [4,5,6]. AMPs can disrupt bacterial cell membranes through the carpet, toroidal-pore, and barrel-stave disruption mechanisms, which is quite different from conventional drugs [7]. In addition to antibacterial activities, many AMPs also have significant anti-inflammatory activities [8,9,10,11,12]. This is because some AMPs can directly interact with LPS, thereby effectively alleviating LPS-induced inflammation [13,14,15]. AMPs that neutralize LPS and inhibit the release of inflammatory cytokines may have potential clinical applications. 

Peptide hybridization is a simple and effective strategy to obtain novel AMPs with higher antibacterial and lower cytotoxicity than parent peptides [16,17,18]. CM4, LL37, and TP5 are the parent peptides for our rational design. The CM4 peptide, which is isolated from the *silkworm Bombyx mori*, is a 35-amino acid-residue small cationic antimicrobial peptide [19]. CM4 has potent antibacterial, antitumor, and antifungal properties by permeabilizing the cell membrane without being toxic to normal mammalian cells [20]. Furthermore, it has the potential to inhibit the release of cytokines and nitric oxide (NO) through direct binding to LPS [21]. This offers the prospect of CM4 being used as a therapeutic candidate for the treatment of sepsis and endotoxic shock. LL37, composed of 37 amino acids, is a cationic antimicrobial peptide found in human leukocytes [13]. Not only has it been reported to play a significant function in bacterial infection defense, but it also interacts directly with LPS to suppress inflammation [22,23,24]. Therefore, LL37 can be developed as a clinical candidate for treating inflammation and reducing the effects of endotoxin. Although LL37 possesses remarkable antibacterial and anti-inflammatory activities, its cytotoxicity to eukaryotic cells severely hinders its clinical development [13]. Thymopentin (TP5), composed of five amino acids, is a polypeptide derived from a partial fragment of thymopoietin [25]. TP5 regulates immunological and inflammatory responses by playing a critical role in T-cell maturation and differentiation [26]. TP5 is commonly used in the clinical treatment of immunodeficiencies [27], rheumatoid arthritis [28], malignancies [25], and acquired immunodeficiency syndrome [29]. However, the relatively weak anti-inflammatory activity of TP5 development has prevented its use for clinical purposes.

The clinical application of AMPs requires an efficient and economical production method. The conventional chemical synthesis of AMPs is a very complex and difficult procedure. The superlative approach to solving this issue is by heterologous expression technology. In our previous work, we expressed a peptide termed CLP fused with a SUMO tag and demonstrated that the purified CLP had antibacterial activity [17]; however, the protein purified from bacteria still requires expensive LPS removal before clinical application. This has led us to seek other expression platforms.

The yeast expression system (*Pichia pastoris*) has noticeable advantages over the prokaryotic expression system, including easy purification, high cell density, and post-translational modifications [30,31]. Many AMPs have been successfully expressed by using the *P. pastoris* system, such as Apidaecin [32], EF-1 [33], and LL37 [34]. Additionally, human serum albumin (HSA) is thought to enhance the secretion of antimicrobial peptides in *P. pastoris* and provide increased stability [31].

In the present study, we hypothesized that the combination of CM4 [1,2,3,4,5,6,7,8], LL37 [13,14,15,16,17,18,19,20,21,22,23,24,25,26,27,28,29,30,31,32,33,34,35,36], and TP5 [1,2,3,4,5] may have amended LPS neutralizing and anti-inflammatory activity with the least cytotoxic and hemolytic effects. Consequently, we synthesized and expressed the hybrid peptide CLP fused with HSA in methylotrophic yeast and studied its natural activities.

## 2. Materials and Methods

### 2.1. Plasmids, Strains, and Reagents

*Pichia pastoris* (strain GS115) and zeocin were obtained from Invitrogen (Carlsbad, CA, USA). Plasmid pPICZαA-Amp was available in our laboratory. Plasmid Miniprep kit and *E. coli* DH5α chemically competent cells were procured from Tiangen Biotech (Beijing, China). Pme Ⅰ, Bsa Ⅰ-HFv2 restriction enzymes, Q5 High-Fidelity DNA Polymerases, and T4 DNA ligase were procured from New England Biolabs (Ipswich, MA, USA). TEV protease was purchased from Solarbio (Beijing, China). LPS from *E. coli* O55: B5 was procured from Sigma (St. Louis, MO, USA).

### 2.2. Construction of Expression Plasmid pPICZαA-HSA-CLP

The gene of *HSA-CLP* (linking CLP peptide and fusion partner HSA with TEV recognition sequence) was optimized and synthesized by Genewiz (Suzhou, China) and cloned into pUC57. To amplify the *HSA-CLP* gene, Primer F (5′-AAAGGTCTCAATCCGATGCACACAAGAGTGAGGTTG-3′) and Primer R (5′-AAAGGTCTCAAGCCTCAGGTAACATCCTTACGTTCAGTACGAG-3′) were designed. PCR conditions were performed at 98 °C for 30 s, then followed by 35 cycles (98 °C/10 s; 55°C/30 s; 72 °C/30 s) and a final extension at 72 °C for 2 min. The amplified products were detected using 1.5% agarose gel and purified using the GenElute™ Gel extraction kit (Omega, Norwalk, CT, USA). The purified product *HSA-CLP* and plasmid pPICZαA-Amp were digested/ligated using the golden gate cloning method to create the pPICZαA-*HSA-CLP* plasmid. The reaction mixture in a 20 µL total reaction volume included 20 ng pPICZαA-Amp plasmid and amplified products, 1.6 µL Bsa Ⅰ-HF v2 restriction enzyme, and 0.4 µL T4 DNA ligase buffer. The reaction conditions were carried out at 37 °C/30 min, then followed by 16 cycles (37 °C/10 min; 16°C/5 min), then followed by 16 °C/60 min, and a final extension at 80 °C/6 min. The products of the golden gate reaction were transformed into *E. coli* DH5α competent cells, which were then spread on Low-salt Luria-Bertani (LB) plates with 100 µg/mL zeocin. After overnight incubation at 37 °C, plate-grown colonies were spread onto LB plates containing 50 µg/mL ampicillin. Positive colonies grew on Low-salt LB plates with 100 µg/mL zeocin but not on LB plates containing 50 µg/mL ampicillin. The recombinant plasmids were extracted and then confirmed by Sanger sequencing.

### 2.3. Expression and Purification of Fusion HSA-CLP Peptide

The pPICZαA-*HSA-CLP* recombinant plasmids were linearized with the Pme I restriction enzyme and electroporated into the *P. pastoris* GS115 chromosome according to the manufacturer’s instructions. The transformed cells were incubated for 48 h at 30 °C on yeast extract peptone dextrose (YPDS) medium (2% peptone, 1 M sorbitol, 1% yeast extract, 2% agar, 2% dextrose, 100 µg/mL zeocin) plates. A total of 10 zeocin-resistant colonies were selected and confirmed by PCR and Sanger sequencing using universal primers 5′ alcohol oxidase 1 (*AOX 1*) and 3′ *AOX 1*. The recombinant yeast cells were cultured in a buffered glycerol-complex (BMGY) medium (1% glycerol, 2% peptone, 0.1 M potassium phosphate, biotin 4 × 10^−5^%, 1% yeast extract, 1.34% YNB, pH 6.0) at 30 °C, 250 rpm until OD_600_ = 1.0. To induce expression, the recombinant yeast cells were centrifuged at 3000× *g* for 10 min and resuspended in buffered methanol-complex (BMMY) medium (0.5% methanol, 2% peptone, 0.1 M potassium phosphate, biotin 4 × 10^−5^%, 1% yeast extract, and 1.34% YNB, pH 6.0). An amount of 0.5% methanol was added every 24 h to induce expression. SDS-PAGE was used to examine the expression of the fusion *HSA-CLP* peptide in the culture supernatant. The fusion *HSA-CLP* peptide was purified with a 5 mL Ni-NTA column (Sangon Biotech, China). Buffer A (5 mM imidazole, 500 mM NaCl and 20 mM Tris-HCl, pH 8.0) was used for column equilibration. The *HSA-CLP* fusion protein contained a His-tag that allowed binding to Ni-NTA columns. The recombinant peptides were then eluted using a gradient concentration of imidazole (100–500 mM). SDS-PAGE was used to analyze the eluted peptides.

### 2.4. Cleavage and Purification of Hybrid Peptide CLP 

The fusion *HSA-CLP* peptide was cleaved by TEV protease at 16 °C for 6 h. The hybrid peptide CLP and the HSA part were then isolated using an Amicon Ultra centrifugal filter with a 10 kDa cutoff membrane (Millipore, MA, USA). The small CLP peptides collected in the flow-through were analyzed using SDS-PAGE. The purified hybrid peptide CLP was analyzed by electrospray ionization-mass spectrometry (ESI-MS). The concentration of the purified hybrid peptide CLP was measured using the bicinchoninic acid (BCA) method.

### 2.5. Activity Assay of Recombinant CLP Peptide 

#### 2.5.1. LPS Neutralization

The chromogenic Limulus amoebocyte lysate (LAL) test was performed to analyze the effect of LPS neutralized by the hybrid peptide CLP and the parent peptides (CM4, LL37, and TP5). One EU/mL LPS was cultured with variable concentrations of the hybrid and parent peptides (0 to 50 µg/mL) in an endotoxin-free 96-well plate at 37 °C. Additionally, then 50 µL of LAL reagents were added to each well and incubated at 37 °C for 30 min. After 100 µL, the aliquots of the chromogenic substrate were incubated at 37 °C for 6 min until a yellow color appeared. A terminator was added to stop the reaction, and the absorbance was analyzed at 405 nm.

#### 2.5.2. Hemolytic Activity

The hemolytic activity of the hybrid peptide CLP was measured using mouse red cells (RBCs). Fresh mouse RBCs were washed twice with 1 × PBS and then diluted to 10% hematocrit. The cells were treated for 1 h at 37 °C with various concentrations of the hybrid peptide and parent peptides (10 to 60 µg/mL). The absorbance of the culture supernatant was analyzed at 414 nm. Triton X-100 and PBS were employed as the positive and negative controls.

### 2.6. Cell Culture

The mouse RAW264.7 macrophages were grown in Dulbecco’s modified Eagle’s medium (DMEM) supplemented with 1% streptomycin, 10% fetal bovine serum, and 1% penicillin. Cells were cultivated in a humidified cell incubator with 5% CO_2_ at 37 °C.

#### 2.6.1. Cell Viability Assay

The Cell counting kit-8 (CCK-8) was used to determine the viability of the peptides that treated IPEC-J2 cells. Cells were diluted to 3 × 10^5^ cells/mL and incubated overnight in a 96-well plate. The hybrid and parent peptides (concentrations ranging from 10 to 60 µg/mL) were then placed in the 96-well plate and cultured for 24 h at 37 °C. A total of 10 µL of CCK-8 reagents were introduced to the 96-well plate and cultured for 2 h in the dark. Then, the absorbance was analyzed at 405 nm with a microplate reader.

##### 2.6.2. LPS-Induced Secretion of Nitric Oxide (NO) in Mouse RAW264.7 Macrophages

Mouse RAW 264.7 macrophages were cultured with various concentrations of the hybrid peptide and parent peptides (10 to 50 µg/mL) for 1 h. The peptide-treated cells were then exposed to 1 µg/mL LPS for 24 h. The culture supernatant of cells was collected and incubated with Griess reagent. A microplate reader was used to measure absorbance at 540 nm.

#### 2.6.3. Evaluation of CLP on LPS-Induced Pro-inflammatory Cytokines in Mouse RAW264.7 Macrophages

Mouse RAW264.7 macrophages were incubated with the hybrid and parent peptides (10–50 µg/mL) before exposure to 1 µg/mL LPS. After culturing at 37 °C for 24 h with 5% CO_2_, the TNFα, IL-6, and IL-1β levels in the culture supernatant were determined by an enzyme-linked immunosorbent assay (ELISA) kit (eBioscience, CA, USA). Samples were collected and analyzed at 450 nm absorbance.

#### 2.6.4. Effects of CLP on LPS-Induced Apoptosis of Mouse RAW264.7 Macrophages

Annexin V-FITC (Fluorescein isothiocyante) and the PI (Propidium iodide) double-staining method were used to detect apoptosis in LPS-stimulated mouse RAW 264.7 macrophages. The mouse RAW 264.7 macrophages were treated with or without the hybrid and parent peptides and incubated for 1 h at 37 °C, followed by exposure to 10 µg/mL LPS for 4 h, 12 h, and 24 h, respectively. The cells were washed three times with 1 × PBS and stained with Annexin V-FITC and PI. Flow cytometry was used to assess the rate of apoptosis.

### 2.7. Statistical Analysis

The data were indicated as the mean ± standard deviation (SD) of three separate experiments. Student *t*-tests and one-way ANOVA were used for statistical comparisons. *p* < 0.05 denotes a statistical difference, *p* < 0.01 denotes a statistically significant difference, and *p* < 0.001 denotes a highly significant statistical difference.

## 3. Results

### 3.1. Construction of Recombinant Plasmid pPICZαA-HSA-CLP

Previously, an effective strategy for the rapid construction of target genes into the pPICZαA-Amp plasmid using the golden gate technology was developed [35]. The schematic diagram of the recombinant plasmid construction is shown in Figure 1. The *HSA-CLP* gene was synthesized and cloned into pPICZA-Amp to create the recombinant plasmid pPICZA-*HSA-CLP*. His tag was attached to the C terminal of HSA. There was a TEV protease cleavage site between HSA and CLP. The constructed plasmid was verified by Sanger sequencing.

### 3.2. Expression and Purification of Fusion HSA-CLP Peptide

The pPICZαA-*HSA-CLP* vector was linearized and transformed into the GS115 strain of *P. pastoris* by electroporation, and the expression of the fusion *HSA-CLP* peptide was induced by adding 0.5% methanol in the BMMY medium. After centrifugation, the fusion peptides were collected and analyzed by SDS-PAGE. As revealed in Figure 2A, a prominent band around 74 kDa was observed, corresponding to the predicted molecular weight. The maximum secreted expression of *HSA-CLP* was observed at 60 h of methanol induction. The His-tagged *HSA-CLP* was purified by the Ni-NTA column. The SDS-PAGE results revealed a single clean band at 74 kDa (Figure 2B, lane 2).

### 3.3. Cleavage and Purification of Hybrid Peptide CLP

To obtain the hybrid peptide CLP, the recombinant protein *HSA-CLP* was processed with TEV protease. The fusion protein was effectively cleaved, and a clear band was exhibited at 5 kDa on SDS-PAGE. (Figure 2B, lane after TEV). The hybrid peptide CLP was purified using an Amicon Ultra centrifugal filter with a 10 kDa cutoff membrane. SDS-PAGE results indicate that the hybrid peptide CLP was successfully separated in the fraction of flow-through (Figure 2C, lane CLP). The highest yield of 101.71 mg/L for the hybrid peptide CLP was obtained. The purified hybrid peptide CLP was further identified by ESI-MS. The exact molecular weight of CLP was 4628.80 Da (Figure 2D), which was close to the predicted value of 4628.62 Da.

### 3.4. Inhibition of LPS Activity and Pro-Inflammatory Cytokine Induction

#### 3.4.1. Recombinant Peptide CLP Inhibitory Effect on LPS

The CLP is a cationic amphiphilic hybrid peptide with a net charge of +9, so we expected CLP to neutralize LPS by binding. The LAL assay is a sensitive method to determine the content of free LPS. This method was used to assess the ability of the hybrid peptide and parent peptides to neutralize LPS in vitro. Our results revealed that the parent peptides (CM4 and LL37; Figure 3A, green and purple bars, respectively) and the hybrid peptide CLP (Figure 3A, blue bars) were able to inhibit LPS activity in a dose-dependent manner. Furthermore, CLP significantly increased the inhibition of LPS activity compared to the parent peptides at 50 μg/mL (Figure 3A, green and purple bars, 50) to almost complete inhibition (97.648%; Figure 3A, blue bar, 50). 

##### 3.4.2. Cytotoxicity and Hemolytic Activity of CLP

The toxicity of CLP and parent peptides toward the IPEC-J2 cells was determined using the CCK-8 assay. IPEC-J2 cells were treated with various concentrations of the hybrid peptide and parent peptides (10 to 60 µg/mL). CLP exhibited less cytotoxicity compared to the LL37 peptide. The IPEC-J2 cells retained more than 92% viability even at 60 µg/mL (Figure 3B, blue bar, 60). In addition, CLP showed less hemolytic activity compared to the LL37 peptide (Figure 3C, blue bar, 60). These data suggested that CLP had lower cytotoxicity and hemolytic activity; therefore, it was suitable for further anti-inflammatory experiments.

#### 3.4.3. CLP Downregulates LPS-Induced Inflammatory Response in Mouse RAW264.7 Macrophages

The LPS neutralization ability of CLP in vitro suggested that it could alleviate LPS-induced inflammatory responses. To further investigate this, we evaluated the influence of CLP on LPS-stimulated NO and pro-inflammatory cytokines, such as the production of TNF-α, IL-6, and IL-1β in mouse RAW264.7 macrophages. Our findings show that LPS markedly boosted NO production in mouse RAW264.7 macrophages compared with the control group (82.638 vs. 13.015 µM), and after treatment with CLP (10–50 µg/mL), the level of NO was decreased significantly (Figure 4A, blue bar, 10–50). Similarly, CLP (10–50 µg/mL) treatment significantly reduced the LPS-stimulated secretion of TNF-α, IL-6, and IL-1β in mouse RAW264.7 macrophages compared with the LPS alone treatment group (Figure 4B–D, red bar, LPS). However, treatment with the hybrid CLP peptide inhibited this increase in a dose-dependent manner. The TNF-α level was significantly (*p* < 0.001) abridged to 306.443 and 183.702 pg/mL at a CLP peptide concentration of 40 and 50 µg/mL, respectively (Figure 4B, blue bar, 10–50). The IL-6 level was also increased with only LPS treatment (339.256 pg/mL), but in the CLP peptide-treated group, the peptide concentration of 30 and 40 µg/mL decreased IL-6 secretion to 201.654 and 171.990 pg/mL, respectively (*p* < 0.001) (Figure 4C, blue bar, 10–50). Correspondingly, the CLP peptide also repressed the secretion of IL-1β to 468.138 and 420.568 pg/mL at the peptide concentration of 20 and 30 µg/mL compared to the LPS-stimulated (577.267 pg/mL) mouse RAW 264.7 cells, respectively (*p* < 0.001) (Figure 4D, blue bar, 10–50). These results indicate that the hybrid peptide CLP has potent anti-inflammatory activity. Furthermore, compared to the parental peptides, the hybrid peptide revealed additional anti-inflammatory activities (Figure 4A–D, purple bar, 10–50).

#### 3.4.4. Effects of CLP on LPS-Induced Apoptosis of Mouse RAW264.7 Macrophages

To study the impact of the hybrid peptide, CLP was compared to parent peptide LL37 on LPS-stimulated apoptosis. The mouse RAW264.7 macrophages were treated with LPS alone or co-treated with LPS and peptides (LL37 or CLP) for 4, 12, and 24 h. The Annexin V-FITC and PI-labeled cells were evaluated by flow cytometry. The results are displayed in Figure 5A,B. In comparison to both the control and treatment groups, the LPS-treated group increased cell apoptosis at 4, 12, and 24 h. However, the LPS+CLP treatment group significantly (*p* < 0.01) reduced apoptosis compared with the LPS or LPS+LL37 treatment groups. Our results indicate that CLP reduced LPS-induced apoptosis in mouse RAW264.7 macrophages by neutralizing LPS.

## 4. Discussion

The emergence and spread of drug-resistant bacteria have become a major global concern [36,37,38]. AMPs are thought to be the natural defense against microbial infections [39]. In addition to their direct antibacterial effects, many AMPs also exhibit a strong affinity for LPS [40,41], inhibit the secretion of pro-inflammatory cytokines and nitric oxide by neutralizing LPS, and help regulate inflammatory responses. Therefore, AMPs are considered virtuous candidates for neutralizing endotoxin activity. In recent years, scientists have modified the amino acid sequence of parent peptides to obtain novel antimicrobial peptides with optimal antibacterial, immunomodulatory, and anti-inflammatory activities [18,42,43]. The conserved amino acid of AMPs is crucial to their biological activity [44]. However, the proper substitution of some conserved sequences improves the activity of the peptide [45]. The hybridization of different parent peptides is an efficient approach to increase anti-inflammatory and antibacterial activities with minimal side effects, such as cecropin [46], LL37 [47], cathelicidin [48], and melittin [49].

There are various methods for producing antimicrobial peptides, including extraction from animals and plants, artificial synthesis, and recombinant expression. Due to the complex extraction process from natural resources and the high synthetic price, the *P. pastoris* recombinant expression system offers a solution for the industrial production of AMPs. In addition, HSA is thought to enhance the secretion of antimicrobial peptides in *P. pastoris* and provide stability [31]. In this study, the hybrid peptide CLP was successfully expressed as a fusion in *P. pastoris*. Compared with the *E. coli* expression system, the *P. pastoris* expression system secretes fewer host cell proteins but a large number of recombinant target proteins into the medium [50]. At the same time, the system contains an α-factor signal peptide that can secrete the recombinant protein outside the cells [51]. These properties greatly reduce the downstream purification burden. Moreover, it is free of endotoxins and viruses [52]. The *P. pastoris* expression system has successfully achieved the heterologous expression of a variety of peptides [53,54,55,56,57]. In this study, after 60 h of methanol (0.5%) induction, we obtained 2.07 g/L of the fusion protein *HSA-CLP*. After purification by the Ni-NTA pre-packed gravity column, digestion with TEV protease, and purification by ultrafiltration through a 10 kDa ultrafiltration tube, 101.71 mg/L of recombinant hybrid peptide CLP was obtained. Its expression level was higher than the previously reported antimicrobial peptides DEFB-TP5 (30.41 mg/L) [11], ABP-CM4 (15 mg/L) [58], and Hispidalin (20.4 mg/L) [59]. SDS-PAGE findings revealed that the molecular weight of recombinant CLP was about 5 kDa. ESI-MS analysis showed a molecular weight of 4628.80 Da, which was consistent with the expected molecular weight.

In the current study, the recombinant hybrid peptide CLP was determined by LPS neutralization, hemolytic activity, and cytotoxicity, which is regarded as a crucial feature of AMPs and for their use as a highly effective antibacterial agent. The LAL is a bacterial lipopolysaccharide (LPS) detection reagent derived from the blood cells of *Limulus polyphemus*, which is widely used in the detection and quantification of Gram-negative bacterial endotoxin [60]. In this study, the LAL test demonstrated that CLP neutralized its endotoxin activity, thereby blocking the endotoxin-induced activation of downstream signaling pathways. Furthermore, the effect on LPS inhibition was better than the previously expressed peptides DEFB-TP5 [12] and CATH-2TP5 [16].

In addition, the parent peptides have antibacterial activity against Gram-negative bacteria but also have hemolytic activity and cytotoxicity against normal mammalian cells [61]. The hybrid peptide CLP consisting of an N-terminal with polar amino acids and a C-terminal with decreased hydrophobic amino acids was able to efficiently neutralize LPS. It supports our hypothesis that this hybrid peptide exhibits stronger anti-inflammatory activities and minimal cytotoxicity compared to the parent peptides and significantly enhances its neutralization capacity to LPS; these findings are consistent with previous research findings [62,63,64,65]. 

In our current study, we observed the effects of CLP on the LPS-stimulated secretion of NO, TNF-α, IL-6, and IL-1β in mouse RAW264.7 macrophages. Endotoxins are produced when bacteria die and disintegrate, which plays a major role in causing septic shock syndrome [66,67,68]. When subjected to external stimuli (such as LPS), macrophages play a part in killing or inhibiting the reproduction of pathogens by releasing stress effector molecules such as NO, which are indispensable regulators for the functioning of the immune system [69]. However, the excessive secretion of NO can lead to different inflammatory reactions, such as septic shock, ulcerative colitis [70], and arthritis [71]. Therefore, the inhibition of NO synthesis might be a novel approach to improving inflammatory diseases. Additionally, macrophages that are stimulated by external factors also secrete a high number of pro-inflammatory cytokines, which may lead to inflammatory diseases such as rheumatoid arthritis, hemorrhagic shock, and atherosclerosis [72]. Therefore, inhibiting the secretion of pro-inflammatory cytokines is crucial for alleviating inflammatory illnesses. Compared with previously described lunasin-4 [73] and SPHF1 [74], CLP has a better ability to suppress the production of pro-inflammatory cytokines.

In previous studies, cationic peptides were reported to interact with the CD14 receptor on mouse RAW264.7 macrophages, thereby inhibiting the binding of LPS to the CD14 receptor [75,76]. This study recommends CLP as an effective peptide that neutralizes the LPS and reduces the risk of shock syndrome caused by the lysis of Gram-negative bacteria. The in vitro study of CLP peptide showed that this hybrid peptide binds with the CD14 receptor present on the surface of mouse RAW264.7 macrophages and inhibits LPS activity. 

LPS is the main substance produced by Gram-negative bacteria that induce cell damage and apoptosis. Therefore, we speculated that the CLP neutralization of LPS could inhibit LPS-induced apoptosis. In our current study, we explored the effect of CLP on apoptosis in LPS-stimulated mouse RAW264.7 macrophages by flow cytometry. The results showed that CLP was able to reduce the apoptosis of macrophages by neutralizing LPS. Generally, these findings designate that CLP is an auspicious peptide that could be a prospective agent for application in the medical industry.

## 5. Conclusions

In the present study, we used the golden gate cloning strategy to construct the expression vector pPICZαA-*HSA-CLP* and successfully expressed the CLP in the *P. pastoris*. The purified CLP exhibited lower cytotoxicity and hemolytic activity than the parent peptide LL37 and exerted its anti-endotoxin, anti-inflammatory, and immunomodulatory properties by neutralizing LPS. CLP could suppress the secretion of pro-inflammatory cytokines and NO in LPS-stimulated mouse RAW264.7 macrophages and reduce the LPS-stimulated apoptosis of mouse macrophages (Figure 6). These findings demonstrate an approach for the recombinant production of CLP in the industry and suggest that CLP may have a potential therapeutic role in the future.

## Figures and Tables

**Figure 1 microorganisms-11-00131-f001:**
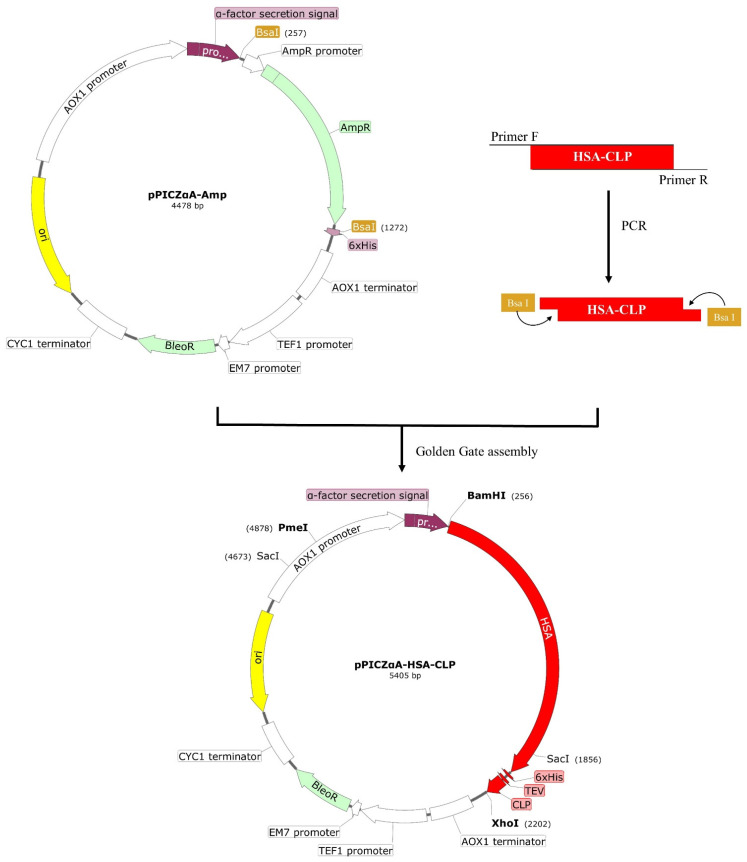
The schematic diagram of the construction expression vector. The target gene was successfully interested with 6 x His and two restriction sites *BamHI and SacI*.

**Figure 2 microorganisms-11-00131-f002:**
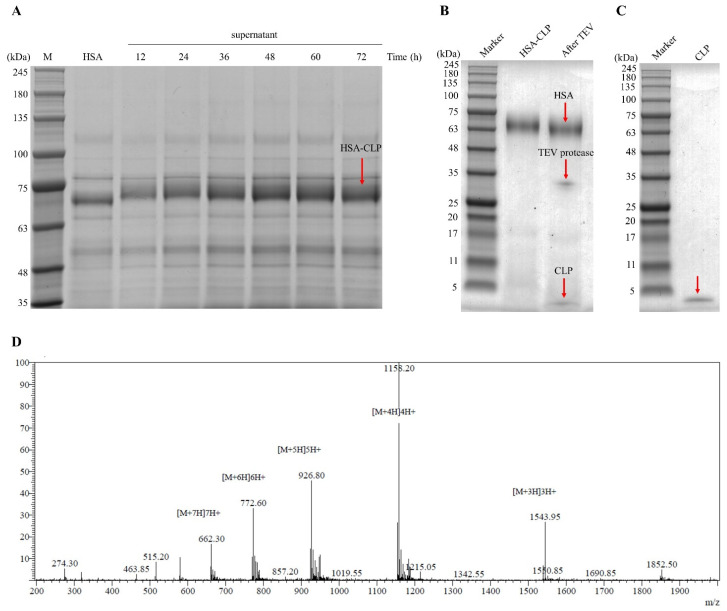
Analysis of recombinant hybrid peptide CLP. (**A**) SDS-PAGE detection of supernatants after methanol induction for 12–72 h (lane M, molecular weight markers; arrow indicates expression of *HSA-CLP*. (**B**) The purified *HSA-CLP* and its cleavage products by TEV protease were detected by SDS-PAGE (marker, molecular weight; arrow shows accepted linear epitope and peptides) (**C**) The purified CLP was detected by the SED-PAGE, marker, molecular weight; arrow shows CLP peptide. (**D**) Analysis of hybrid peptide CLP by ESI-MS.

**Figure 3 microorganisms-11-00131-f003:**
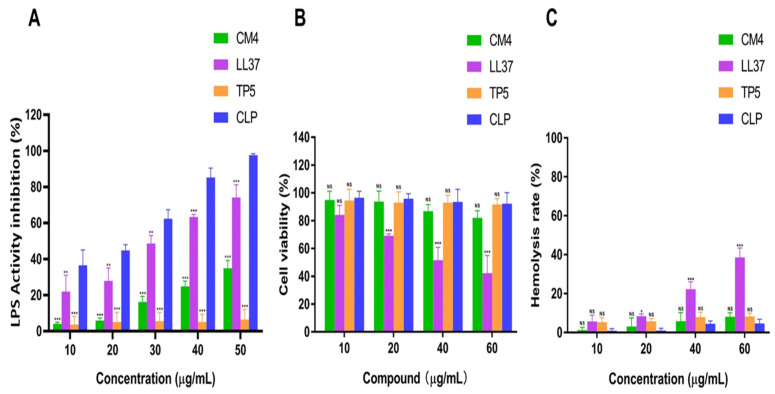
Effects of hybrid peptide CLP and parent peptides (CM4, LL37, and TP5) on LPS activity inhibition, cell viability, and hemolytic activity. (**A**) LPS activity inhibition by CLP and the parent peptides was determined by the LTA test. ** *p* < 0.01 and *** *p* < 0.001 represented a contrast of parent peptides to CLP. (**B**) CLP peptide attenuated the cytotoxicity of LPS on IPEC-J2. Cells NS and *** *p* < 0.001 represented a comparison of parent peptides to CLP. (**C**) Hemolytic activities of CLP and parent peptides against mouse RBCs. NS, * *p* < 0.05, and *** *p* < 0.001 indicated an appraisal of parent peptides vs. CLP.

**Figure 4 microorganisms-11-00131-f004:**
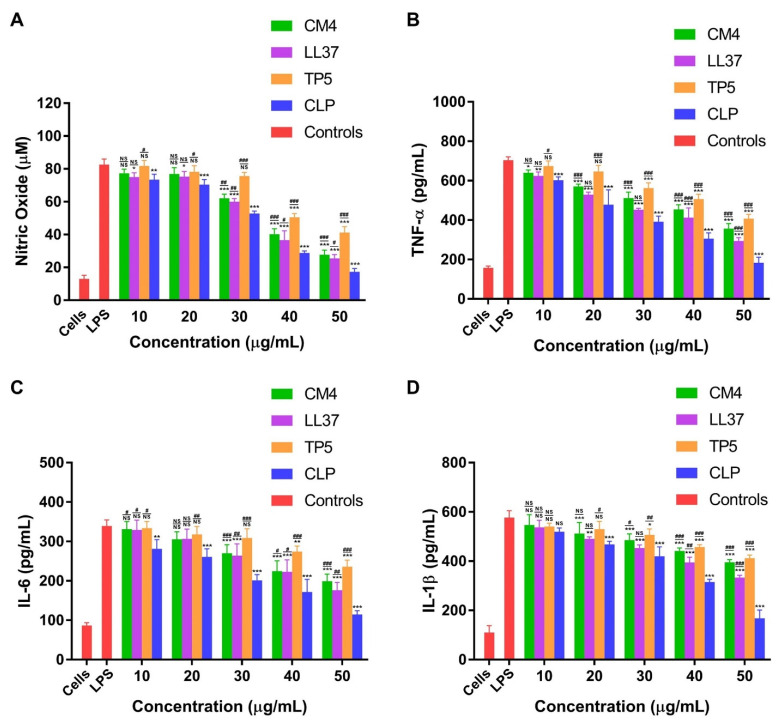
Effect of CLP and parent peptides (CM4, LL37, and TP5) on LPS-stimulated pro-inflammatory cytokine secretion in mouse RAW264.7 macrophages. (**A**) The levels of NO. (**B**) The levels of TNF-α. (**C**) The levels of IL-6. (**D**) The levels of IL-1β. Inflammatory production of cells was induced with LPS and attenuated by treatment with various concentrations of CLP and parent peptides (CM4, LL37, and TP5). The supernatant was collected after 24 h of incubation and the levels of inflammatory cytokines were determined. NS, * *p* < 0.05, ** *p* < 0.01, and *** *p* < 0.001 represented a comparison of the hybrid peptide CLP and parent peptides vs. LPS. NS, ^#^
*p* < 0.05, ^##^
*p* < 0.01, and ^###^
*p* < 0.001 indicated a comparison of the parent peptides vs. CLP.

**Figure 5 microorganisms-11-00131-f005:**
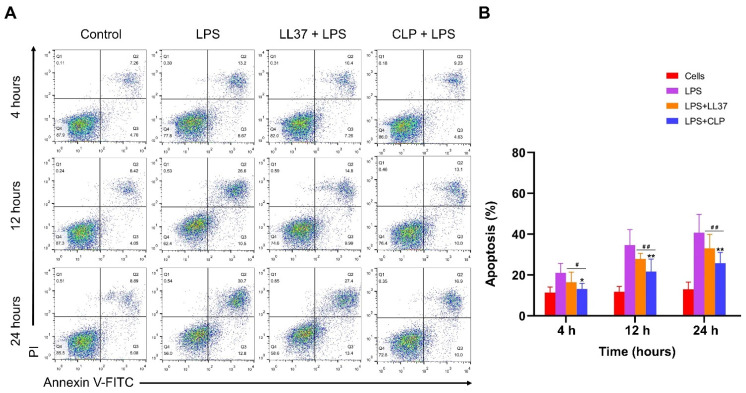
Effects of the parent and hybrid peptides on LPS-induced apoptosis in mouse RAW264.7 macrophages. LPS-induced mouse RAW264.7 macrophages were treated with or without CLP and the parent peptide LL37 at 4, 12, and 24 h. (**A**) Flow cytometry was used to study apoptosis in mouse RAW264.7 macrophages. Normal cells were used as controls, LPS represents cells that have only been treated with LPS; LL37+LPS and LPS+CLP represent cells co-treated with LPS and peptides. (**B**) The apoptosis rate of mouse RAW264.7 macrophages treated with LPS alone or co-treated with peptides. * *p* < 0.05 and ** *p* < 0.01 represented the comparison of LPS+CLP vs. LPS. ^#^
*p* < 0.05 and ^##^
*p* < 0.01 indicated the comparison of LPS+LL37 vs. LPS+CLP.

**Figure 6 microorganisms-11-00131-f006:**
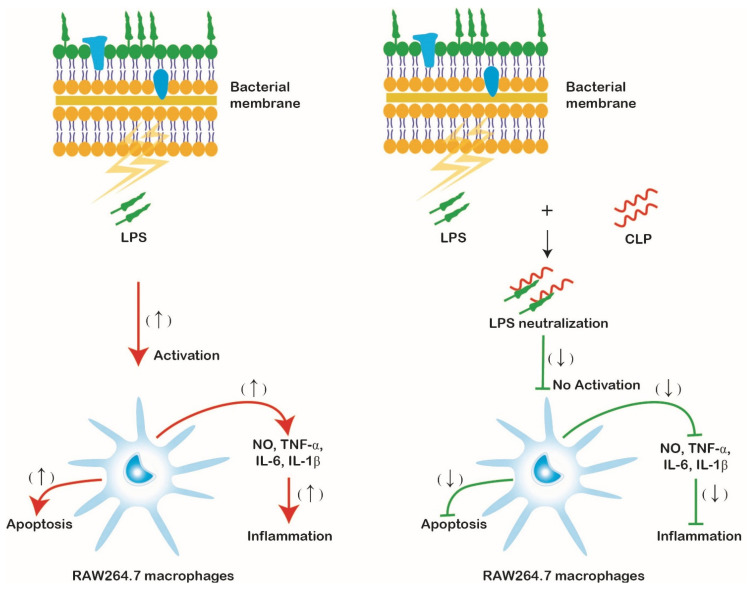
The schematic illustration of the possible mechanism by which CLP inhibits LPS-stimulated inflammatory responses in mouse RAW264.7 macrophages. Right, LPS is neutralized by CLP; left, LPS alone. (↑) indicates up-regulated response; (↓) indicates down-regulated response.

## Data Availability

All concerned data have been provided in the manuscript. There is no supplementary data.

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
