# Peer review of "Yeast Expressed Hybrid Peptide CLP Abridged Pro-Inflammatory Cytokine Levels by Endotoxin Neutralization"

_microorganisms, 2023, doi:10.3390/microorganisms11010131_

Round 1

Reviewer 1 Report

The article is deal with the novel hybrid peptide CLP abridged inflammation by endotoxin neutralization. The topic discussed is very important for the treatment and prevention of infectious diseases.

I would like to make a few comments:

1)                There are mistakes in the text and missing words. For example:

-         The title of the article is ” A Yeast Expressed Novel Hybrid Peptide CLP Abridged Inflammation by Endotoxin Neutralization”. It is better to write: “Yeast Expressed Novel Hybrid Peptide CLP Abridged Inflammation by Endotoxin Neutralization”

-         line 15: What does it mean: “natural application”? or ”natural activities”? Perhaps, it is better to write without word “natural”?

-         line 22: Lipopolysaccirdes (LPS). Correct is: Lipopolysacchirdes (LPS)

-         line 72: word extravagant better to replace on: wasteful

-         line 248: towards IPEC-J2

It is better to write: IPEC-J2 cells or IPEC-J2 cell line

-         line 249: IPEC-J2 was treated

it is better to write: IPEC-J2 cells were treated

-         line 259: IPET-J2. It is a mistake. Correct: it should be: IPEC-J2

-         line 252: In addition, CLP showed less hemolytic compared to the LL37 peptide

It should be: In addition, CLP showed less hemolytic activity compared to the LL37 peptide

2)                In Materials and Methods:

-         lines 181, 187: Specify the sensitivity threshold of Griess reagent and ELISA kits

3)                It is known that stability of antimicrobial peptides in blood is very low. Proteins of blood serum neutralize their activity in a great extent.   All experiments in this article were in vitro studies.   It is necessary to indicate in Discussion the limitations of the studies performed, in particular, the lack of data on interaction with blood plasma proteins. It is desirable in the Discussion, that the authors put forward a hypothesis explaining the decrease in the cytotoxic activity of the hybrid protein containing two cytotoxic antimicrobial peptides.

Author Response

Thanks to the reviewer for the valuable suggestion and comments.

For lines 15,22, 72,248,249, and 252,259, 181,187 suggestions and changes have been done. 

The introduction, methods, result, and conclusion part has improved according to your suggestion.

Reviewer 2 Report

The manuscript entitled “A Yeast Expressed Novel Hybrid Peptide CLP Abridged Inflammation by Endotoxin Neutralization” by Cheng et al., presents the assembly of three antimicrobial peptides into a compound polypeptide that was expressed in Pichia pastoris, purified, and tested for several activities. The activities found included LPS activity neutralization, and inhibition of NO and pro-inflammatory cytokine levels.  The experiments are well performed but several grammatical mistakes and inappropriate choice of words were also found. In addition, some conclusions are not supported by the presented data, and several other issues must be corrected before it is suitable for publication in “Microorganisms”.

1. Abstract.

-Lines 2-3. Inappropriate title: “abridged inflammation” was not actually tested and “abridged pro-inflammatory cytokine levels” is what was observed; also avoid the use of the term “novel”. A more appropriate title is “A Yeast Expressed Hybrid Peptide CLP Abridged Pro-inflammatory Cytokine Levels by Endotoxin Neutralization”.

-Line 14. I think the authors meant “The aim of the study” rather than “The amid…”

-Lines 14-15. The terms “to construct the strategy” and “natural application” are not appropriate. It is better to say “The aim of the study was to apply a strategy to express a recombinant CLP peptide and explore its application as a product derived from natural compounds.”

-Line 22. “Lipopolysaccirdes” must be corrected to “lipopolysaccharides” without initial capital letter.

-Lines 21-22. I am not sure the term “neutralized lipopolysaccharides” is applicable. I think “inhibited lipopolysaccharide (LPS) endotoxin activity” is more appropriate.

-Line 23. “proficient” should be replaced by “efficient”.

-Lines 25-26. Avoid using dangling sentences as “As an attendant to control and the parental peptide LL37, the number of LPS-induced apoptotic cells was also contracted..”, and be more concise. For example: “The number of LPS-induced apoptotic cells diminished compared to the control parental peptide LL37”.

2. Introduction.

-Lines 55-54. Specify in the sentence “by permeabilizing the cell membrane without being toxic to normal mammalian cells”, which “cell membrane” is permeabilized; Is it the bacterial, fungal or what kind of cell membrane?

-Lines 64, 70. The sentences “severely hinders its clinical development” are used twice in the same paragraph and in both cases their context is unclear; specify that it is the “development of its use for clinical purposes” is what is severely hindered.

-Line 72. The word “extravagant” in the sentence “The conventional chemical synthesis of AMPs is extravagant.”, is inappropriate; it must be changed to “a very complex and difficult procedure”.

-Line 73. The term “to elucidate this issue” is also confusing; use “to solve this issue”.

-Line 74. Change “we expressed the peptide CLP” to “we expressed a peptide termed CLP”.

-Line 75. Change “…antibacterial activity (17), however,” to “…antibacterial activity (17); however,”.

-Line 76. Change “This leads us…” to “This led us…”

-Lines 78-79. Change “advantages associated with the prokaryotic expression system” to “advantages over the prokaryotic expression system”.

-Lines 84-85. Are the ranges of reference numbers correct? “CM4 (1-8), LL37 (13-36), and TP5 (1-5)” do not seem to correspond and the ranges are huge.

3. Materials and Methods.

-Line 99, 101, 107, 108, 121. The gene name “HSA-CLP” should be in italics.

-Line 125, 173. Use “Ten” instead of “10” at the beginning of the sentence.

-Line 145. There is no mention of how the proteins on the gels were detected. The addition of the staining protocol is necessary.

-Line 152. Use “One” instead of “1” at the beginning of the sentence.

-Lines 154-156. Unclear sentence “After 100 μL aliquots of the chromogenic substrate were incubated at 37°C for 6 minutes until yellow color appears.”.

-Line 191. Define FITC and PI.

4. Results.

-Line 206. Change “publicized” to “shown”.

-Lines 206-209, 214. The gene name “HSA-CLP” should be in italics.

-Line 212. Figure 1 legend is scarce; a much more detailed description with the main elements of the constructs must be provided.

-Line 219. Add the figure, lane, arrow, etc. to which the result is referred to.

-Line 221. There is no lane 2 in figure; use the lane label; i.e. HSA-CLP.

-Lines 223-226. Poor description of the figure legend. Add detail on the main observations from each lane. Line 225. What is SED-PAGE? 

-Line 229. The sentence “SDS-PAGE investigation appeared that…” is unclear.

-Line 230. There is no lane 3 in figure; use the lane label; i.e. After TEV, red arrow labeled CLP.

-Line 233. Which lane in figure 2C? What does “highest titer” mean here? Do you mean the highest obtained yield? Make the meaning clear please.

-Line 237. The subheading “Anti-inflammatory and Immunomodulatory of Hybrid Peptide CLP” is misleading and not properly written. No anti-inflammatory or immunomodulatory activity were determined. The subheading must be written to describe exactly what was assayed which was inhibition of LPS activity and pro-inflammatory cytokine induction.

-Line 238. Change “neutralizes the” to “inhibitory effect on”.

-Lines 239-240. The authors state “we expected CLP to neutralize LPS by binding.” but no assay to demonstrate CLP-LPS binding was carried out so it can only be an assumption; thus, this must be clearly stated in the text. All “neutralization” terminology must be changed to “LPS activity inhibition” (Figure 3A Y-axis and lines 240, 242-245).

-Lines 242-246. Poor descriptions of results in the text; clearly refer to which figure, color of bar, the result refers to; use as example: “Our results revealed that the parent peptides (CM4 and LL37; Figure 3A, green and purple bars, respectively) and the hybrid peptide CLP (Figure 3A, blue bars) were able to inhibit LPS activity in a dose-dependent manner. Furthermore, CLP significantly increased the inhibition of LPS activity compared to the parent peptides at 50 μg/mL (Figure 3A, green and purple bars, 50), to almost complete inhibition (97.648%; Figure 3A, blue bar, 50).”.

-Lines 248, 249. Write clearly that IPEC-J2 are cells; for example: “The toxicity of CLP and parent peptides towards IPEC-J2 cells was determined.”. Change “IPEC-J2 was treated” to “IPEC-J2 cells were treated”.

-Lines 250-252 and 267-284. Poor description in the text when referring to the figures, bars, labels, etc.; use example above.

-Line 259. “IPET-J2” should be “IPEC-J2”. 

-Line 273. Change “treatment with hybrid CLP peptides terminated this increase” to “treatment with the hybrid CLP peptide inhibited this increase”.

-Line 274. Delete “emission”.

-Lines 275, 277, 280. Why do the authors use the values of 40 and 50, 30 and 40, and 20 and 30 ug/mL for the different cytokines assayed? Why not refer to the maximum values tested of 40 and 50 for all of them?

-Line 295. Change “impact of the hybrid peptide CLP and parent peptide LL37” to “impact of the hybrid peptide CLP compared to parent peptide LL37”.

-Lines 299-301. The result that mentions “the LPS-treated group increased cell apoptosis at 4, 12, and 24 h. However, the LPS+CLP treatment group significantly reduced apoptosis compared with the LPS or LPS+LL37 treatment groups” is true only for the comparison between LPS+CLP and LPS; there was no statistical significance between LPS+CLP and LPS+LL37; thus, this must be clearly stated in this section.

4. Discussion.

-Line 322. The sentence “The conserved amino acid of AMPs is crucial…” is unclear. Does it refer to a particular conserved amino acid? If so, which one? Or does it refer to a conserved stretch of amino acids? Or the whole sequence is conserved? Make a clear statement about this.

-Line 355. Change “the LPS neutralization aptitude is more advanced” to “the effect on LPS inhibition is better”.

-Line 382, 383-384. The sentences “This study initiated that CLP could also avert” and “This recommends that CLP is a promising therapy” are unclear; please make a clear statement of what you mean.

-In lines 75-76 of the introduction the authors mention “the protein purified from bacteria still requires expensive LPS removal before clinical application”; however, no mention on the costs involves in the production of the CLP peptide is made in the discussion. Either provide a discussion on this part, or delete the mention from the intro. 

5. References.

-Lines 438, 588. Abbreviate journal title.

-Lines 454, 572. Change “DOI” to “doi”.

-Lines 460-461, 465. Pseudomonas Aeruginosa in italics and “aeruginosa”.

-Lines 462-463. Aspergillus Niger in italics and “niger”.

-Line 486. Period at the end.

-Line 514. Candida in italics.

6. Figures.

-Figures 3A, B and C. The X axes need more detail; “concentration” of what? Why does B have “compound” instead?

Author Response

Thanks to the reviewer for the valuable suggestion and comments.

  1. Abstract

Point- Lines 2 to 26

All comments have been addressed with track changes in the submitted manuscript.

  1. Introduction

Lines 54 to 85

All comments have been addressed with track changes in the submitted manuscript.  Antimicrobial peptides (AMPs) selectively kill bacteria/fungi by disrupting their cell membranes and are promising compounds to fight drug-resistant microbes.

  1. Materials and Methods

The gene name “HSA-CLP” should be in italics have been done.

Lines 99 to 191 suggestions have been done.

  1. Results

For lines 206 to 301, all changes have been done.

  1. Discussion

Lins 322 to 384 comments addressed.

The introduction has been improved and added part of the discussion.

  1. References

All changes have been done.

 Figure 6. we just made it to understand the mechanism of peptide function in graphical abstract.

The conclusion and result part has improved according to your suggestion.